



# Investigation of soot precursor molecules during inception by acetylene pyrolysis using reactive molecular dynamics

Anindya Ganguly[1], Khaled Mosharraf Mukut[2], Somesh Roy[2], Georgios A. Kelesidis[3], Eirini Goudeli[1]

[1]Department of Chemical Engineering, The University of Melbourne, Parkville, Victoria, Australia
[2]Department of Mechanical Engineering, Marquette University, Milwaukee, WI, USA
[3]Faculty of Aerospace Engineering, Delft University of Technology, Delft, The Netherlands

*Correspondence to*: Eirini Goudeli (eirini.goudeli@unimelb.edu.au)

**Abstract.** Soot inception by acetylene pyrolysis at 1350-1800 K is investigated using reactive molecular dynamics. The composition and chemical structure of soot precursor molecules formed during inception are elucidated. During soot inception, increasing the process temperature leads to faster depletion of $C_2H_2$ molecules and faster formation of $C_2H_3$, $C_2H_4$, $C_2H_6$, $CH_4$ and $C_2$ with the concurrent appearance of $H_2$ molecules. Small molecules consisting of 1 to 5 C atoms ($C_1$-$C_5$) are formed due to reactive collisions and grow further to larger hydrocarbon compounds consisting of 6-10 C atoms. At initial stages of inception, prior to the formation of incipient soot, 3-member rings are formed, which are associated with the formation of compounds with less than 10 C atoms. Once incipient soot is formed, the number of $C_1$-$C_{10}$ compounds and the number of 3-member rings drops, while the number of 5- and 6-member rings increases, indicating that the formation of larger rings is associated with the growth of soot clusters. The chemical structure of soot precursor molecules obtained by bond order analysis reveals that molecules with up to 10 C atoms are either linear or branched aliphatic compounds or may contain 3-member rings fused with aliphatic components. Molecules with more than 10 C atoms often exhibit structures composed of 5- or 6-member C rings, decorated by aliphatic components. The identification of molecular precursors contributing to soot inception provides crucial insights into soot formation mechanisms, identifying potential pathways of soot formation during combustion.

## 1 Introduction

Soot is formed during incomplete combustion or pyrolysis of hydrocarbons (Michelsen et al. 2020) and exhibits adverse effects on human health (Anenberg et al. 2012), leading to respiratory diseases (Shiraiwa et al. 2017) and premature deaths (Giannadaki, Lelieveld and Pozzer 2016). Soot is the largest contributor to global warming after



$CO_2$ (Bond et al. 2013), affecting local, regional and global climate (Ramanathan et al. 2001). Formation of soot is believed to take place by gas-phase condensation or reaction of precursor molecules and polycyclic aromatic hydrocarbons (PAHs) (Johansson et al. 2018) leading to incipient soot (Wang and Chung 2019) (inception), which grows further by condensation of gas-phase radicals on its surface (surface growth) (Michelsen et al. 2020). These

primary soot nanoparticles collide with each other forming aggregates that can break upon oxidation (Naseri et al. 2022). Even though soot growth by coagulation is rather well understood through advances in numerical modeling (Kazakov and Frenklach 1998, Maricq 2007, Sun, Rigopoulos and Liu 2021), simulations (Kelesidis, Goudeli and Pratsinis 2017a, Kelesidis, Goudeli and Pratsinis 2017b, Kelesidis and Goudeli 2021) and measurements (Maricq 2007, Rissler et al. 2013, Maricq 2014), development of an accurate physical

representation of the early stages of soot formation, and particularly inception (Irimiea et al. 2019), possess significant challenges to kinetic modeling of soot (Appel, Bockhorn and Frenklach 2000), as it requires knowledge of the chemical reaction pathways from gaseous species to soot clusters (Thomson 2023). Thus, a more detailed molecular-level understanding of the species formed during soot inception is essential for the accurate description of those chemical reaction pathways and for the design of selective, soot-free chemical processes.

Kinetic models (Frenklach and Wang 1991, Frenklach and Wang 1994) often assume that soot inception occurs through physical PAH dimerization. This is in contrast to theoretical calculations (Houston Miller, Smyth and Mallard 1985, Miller 1991, Schuetz and Frenklach 2002) revealing much lower PAH concentrations than those of small soot particles (Houston Miller et al. 1985) with short lifetimes (< 75 ps) for PAH dimers smaller than 800 amu (Schuetz and Frenklach 2002, Miller 1991). Such theoretical calculations are consistent with

measurements (Sabbah et al. 2010) of the kinetics of pyrene dimerization, corroborating that physical PAH dimerization does not contribute significantly to soot inception. Recent research on combustion has been focusing on chemical dimerization of PAHs through the formation of covalent bonds (Johansson et al. 2017), with experiments suggesting that acetylene ($C_2H_2$) or vinyl ($C_2H_3$) addition via radical chain reaction leads to molecular growth of PAHs via chemisorption (Johansson et al. 2018). Additionally, shock tube experiments of $C_2H_2$ and

$C_4H_2$ pyrolysis (Kiefer et al. 1992) and *ab initio* simulations of $C_2H_2$ (Zádor, Fellows and Miller 2017) have shown that pyrolysis of aliphatic hydrocarbons produce $C_2H_x$ radicals/intermediates, such as acetylene/ethynyl ($C_2H_2/C_2H$) and ethylene/vinyl ($C_2H_4/C_2H_3$) (Tanzawa and Gardiner 1980) at high concentrations, which either initiate or accelerate the formation of a wide variety of products including cyclopentaring-fused PAHs (Shukla 2012). Density functional theory calculations have shown that acetylene-acetylene reaction via vinylidene

formation leads to methylene cyclopropene, a 3-member ring structure that is rapidly converted into aliphatic isomers (Zádor et al. 2017), indicating a resonance-stabilized hydrocarbon radical chain reaction pathway



(Johansson et al. 2018). These findings suggest that reactions of acetylene are involved in nearly all hydrocarbon fuel pyrolysis processes (Liu et al. 2021) and pure acetylene pyrolysis serves as the basis for understanding pyrolysis of other hydrocarbons to soot formation (Slavinskaya et al. 2019).

Reactive molecular dynamics (MD) simulations based on ReaxFF force field (Castro-Marcano et al. 2012) provide insight into the dynamic formation of soot (Goudeli 2019) and have been used to investigate soot inception (Mao, van Duin and Luo 2017, Yuan et al. 2019, Han et al. 2017) and growth (Yuan et al. 2019) using PAHs as the starting nucleating species. Such simulations (Mao et al. 2017) revealed that above 2000 K, PAHs grow into soot particles via chemical reactions, while below the boiling/sublimation temperature of the nucleating PAHs, soot inception occurs via physical PAH dimerization. The presence of large PAHs (> 398 amu) facilitates physical inception of smaller PAHs (< 202 amu) at low temperature (1000 K) but such PAHs dissociate at high temperature (> 1600 K), where soot clusters are formed by radical-radical reactions (Yuan et al. 2019). Arvelos et al. (Arvelos, Abrahão and Eponina Hori 2019) further demonstrated that at ~2165 K cyclohexanone undergoes molecular decomposition, leading to ethene and ethenone formation, marking a shift from stable product formation to radical-dominated process. Above 1500 K, addition of $C_2H_2$ leads to formation of bridges among PAHs and other unsaturated aliphatic hydrocarbons that compose the soot clusters (Yuan et al. 2019). Liu et al. (Liu et al. 2020) expanded on this by investigating n-decane pyrolysis by ReaxFF simulations above 2300 K, showing that shoot growth accelerates especially above 3000 K, where complete soot particle development and graphitization occur.

At 3000 K, reactive collisions of a multicomponent fuel revealed the formation of aliphatic polyyne-like chains that cyclized to form large rings, while internal bridging between macrocyclic carbon atoms led to PAHs with aliphatic side chains (Han et al. 2017). Also, PAH-like molecules coalesce via a ring closing mechanism, leading to conversion of 5- or 7-member rings to 6-member ones (graphitization) with 3-member rings as intermediates (Han et al. 2017). Similarly, Zhang et al. (Zhang et al. 2023) found that at 3000 K, 2,5-dimethylfuran (DMF) pyrolysis promotes rapid polycyclic aromatic hydrocarbon (PAH) growth, driven by increased dehydrogenation and active site availability. Sharma et al. (Sharma et al. 2021) observed that at low temperatures (< 1200 K), soot nanoparticles obtained by $C_2H_2$ pyrolysis are more prone to coalescence than at higher temperature, consistent with PAH-based clusters (Hou et al. 2022) due to low aromatic-to-aliphatic and C/H ratios. Most of these reactive MD studies of soot or carbon black formation, however, are limited to high temperature regions (> 1800 K) and usually consider common PAH molecules as precursors to soot formation (Zhao et al. 2020). So, the detailed molecular structures of soot precursors contributing to the chemical pathway to soot

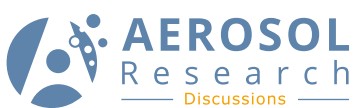

formation have not been thoroughly investigated, particularly at lower temperatures more relevant to combustion conditions.

Here, inception and growth of incipient soot is studied during pyrolysis using acetylene as initial fuel. This study investigates the detailed chemical structure of a wide range of soot precursors, providing a more

comprehensive analysis by accounting for detailed bond information obtained from ReaxFF MD simulations during acetylene pyrolysis at 1350-1800 K. The temporal evolution of the concentration of the most abundant species formed during inception is elucidated along with the average C/H ratio of small (composed of less than 5 C atoms), intermediate (6-10 C atoms) and large molecules (> 10 C atoms) until the formation of super critical nuclei with up to 87 C atoms. The detailed chemical structure of the soot precursors at different stages of soot

inception and surface growth is quantified based on bond order analysis and the effect of temperature on the concentration and structure of these precursors is investigated.

**2 Theory**

**2.1 Inception Simulations**

One thousand acetylene ($C_2H_2$) molecules are randomly distributed in a cubic simulation cell of 75.6 Å length with periodic boundary conditions, using MAPS 4.3 . The use of 1000 $C_2H_2$ molecules ensures sufficient statistical sampling of the population of precursors formed during pyrolysis. The simulation box size was chosen such that 1000 acetylene molecules achieve a density of 0.1 g/cm³ within the simulation domain. This density is consistent with that employed in other ReaxFF studies (Sharma et al. 2021), ensuring that incipient soot formation can occur

within the timescale accessible by reactive MD simulations for the temperature range used in this study. Bond breakage and new bond formation upon molecular collisions is simulated by employing the reactive force field of Castro-Marcano et al. (Castro-Marcano et al. 2012) for hydrocarbons, which has demonstrated consistency with experimental observations (Rokstad, Lindvaag and Holmen 2014, Agafonov et al. 2015, Aghsaee et al. 2014) and theoretical models (Gao and Tang 2022, Saggese et al. 2014, Slavinskaya et al. 2019, Liu et al. 2021), particularly

in understanding radical chain mechanisms in acetylene pyrolysis. The bond lengths of the molecules are constantly adjusted based on their changing local chemical environment (Chenoweth, van Duin and Goddard 2008). ReaxFF enables the simulation of chemically reactive systems [28, 29] through the prediction of atomic connectivity through interatomic distances, angles, and torsion terms [30]. The total system energy is divided into various partial contributions, including bond energy ($E_{bond}$), over-coordination energy penalty ($E_{over}$), under-

coordination stability ($E_{under}$), valence angle energy ($E_{val}$), lone pair energy ($E_{lp}$), penalty energy term ($E_{pen}$), torsion angle energy ($E_{ta}$), conjugation effects to molecular energy ($E_{conj}$), van der Waals energy ($E_{vdW}$), and



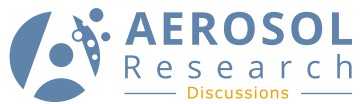

Coulomb energy ($E_{Coul}$):

$$E_{system} = E_{bond} + E_{over} + E_{under} + E_{val} + E_{lp} + E_{pen} + E_{tors} + E_{conj} + E_{vdW} + E_{Coul} \qquad (1)$$

The equations of motion are integrated using a velocity-Verlét algorithm (Swope et al. 1982) with a timestep of 0.25 fs (Chenoweth et al. 2009), consistent with ReaxFF MD simulations (Mao et al. 2017, Lümmen 2010, Rom et al. 2013) of hydrocarbon reactions for soot formation at low temperature, using the Nosé-Hoover thermostat (Evans and Holian 1985) with a damping parameter of 10 fs. Acetylene pyrolysis simulations are carried out for 10 ns in the NVT (constant number, volume and temperature) ensemble using LAMMPS (Plimpton

1995) at 1350, 1500, 1650 and 1800 K. The employed temperature range is consistent with temperature measurements during pyrolysis of ethylene (Dewa et al. 2016, Mei et al. 2019) and acetylene (Drakon et al. 2021). The simulation results were reproduced with up to three additional NVT inception simulations with different initial $C_2H_2$ configurations at each temperature (Supplementary Information). The ReaxFF MD simulation details are listed in Table 1.

**Table 1: MD simulation parameters**.

| | |
|---|---|
| MD integration timestep | 0.25 fs |
| Initial fuel ($C_2H_2$) density | 0.1 g/cm$^3$ |
| Simulation cell dimensions | 75.6 x 75.6 x 75.6 Å |
| Pyrolysis temperature | 1350-1800 K |
| Total duration of pyrolysis simulations | 10 ns |
| Thermostat damping constant | 10 fs |

**2.2 Chemical structure of soot precursor molecules**

The chemical structure of all species consisting of up to 87 C atoms formed during pyrolysis and their number

concentration is obtained as a function of time and is recorded every 0.25 ns. Specifically, the number of bonds between each pair of C-C atoms is calculated by combining bond order information with the octet rule (Lewis 1916). The 3-, 5- and 6-member ring structures are distinguished from non-cyclic ones for each molecule at different timesteps by analyzing the coordinates and bond order of all atoms and their neighbors in each molecule. The aromatic rings are identified by applying Hückel's rule (Solà 2022), accounting only for integer bond order

(single, double and triple bonds) (Balaban and Randić 2004). The bond order values are correlated with the bond



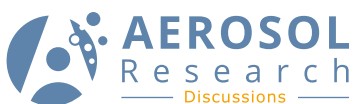

lengths, where for single C-C bond the bond order is < 1.33 (Emri and Lente 2004), for double C-C bond the bond order values range between 1.33-2.85 (Emri and Lente 2004, Hermann and Frenking 2016), and for triple C-C bond the bond order is at least 2.86, corresponding to the triple C-C bond in $C_2H_2$ (Hermann and Frenking 2016). A bond order greater or equal to 1.33 (graphite) (Emri and Lente 2004) indicates the presence of a double bond

for aromatic compounds and bond order of ~1.92 (ethylene) indicates double bonds in aliphatic compounds (Hermann and Frenking 2016). The detailed chemical structure of each individual molecule is visualized using ChemTraYzer (Döntgen et al. 2018), which utilizes the bond order information and the coordinates of each atom in that molecule. The snapshots of the simulated system were visualized using VMD (Humphrey, Dalke and Schulten 1996) and the detailed chemical structures of the molecules are visualized with MolView (Bergwerf

2015) using the simplified molecular-input line-entry system (SMILES) codes generated by the ChemTraYzer software.

### 3 Results and Discussion

**3.1 Molecular properties of soot precursors**

Figure 1 shows snapshots of the growth of carbonaceous nanoparticles formed by acetylene pyrolysis at (a) 1350, (b) 1500, (c) 1650 and (d) 1800 K at times, $t$ = 0.75, 2, 3.75, 5, 6 and 8 ns. All C-C bonds are represented by black lines and all H atoms are omitted for clarity. Initially ($t$ = 0.75 ns), at low temperatures (1350 and 1500 K), acetylene molecules hardly react with each other. At 1350 K, a few linear molecules appear at 2 and 3.75 ns (Fig.

1a: red-circled molecules) due to the reactive collisions of acetylene molecules, which grow into cyclic hydrocarbons at 5 ns (Fig. 1a: green-circled molecules) and later ($t$ = 6 ns) into the incipient soot (Fig. 1a: blue-circled cluster). At 1500 K, linearization is observed at 2 ns (Fig. 1b: red-circled molecules), while cyclization takes place earlier than 1350 K, at 3.75 ns (Fig. 1b: green-circled clusters). After cyclization, the chain and cyclic molecules grow rapidly, forming a large incipient soot nanoparticle at 6 and 5 ns for 1350 and 1500 K,

respectively, by scavenging the surrounding reactive molecules around the cluster (Fig. 1a and b: blue-circled clusters). These clusters grow further by surface growth (8 ns for both 1350 and 1500 K) until most of the surrounding reactive molecules are depleted.



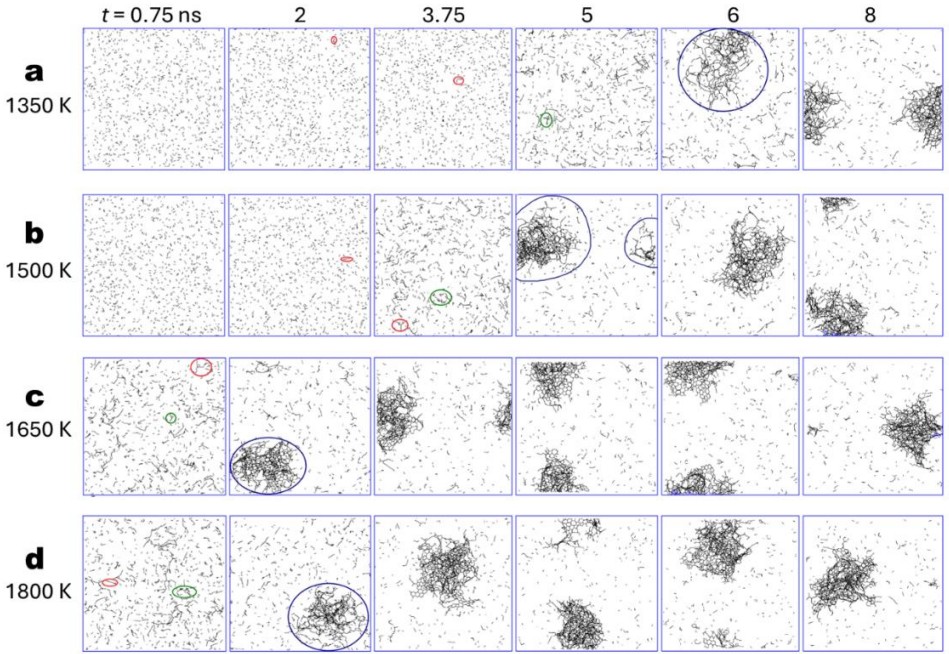

**Figure 1: Snapshots of the growth of carbonaceous species (hydrogen atoms are omitted) formed by acetylene pyrolysis at (a) 1350, (b) 1500, (c) 1650 and (d) 1800 K at times, $t$ = 0.75, 2, 3.75, 5, 6 and 8 ns. Formation of linear-like (chain) molecules (red-circled molecules) is observed early on, especially at high temperatures, followed by the formation of cyclic hydrocarbons (green-circled molecules). These linear and cyclic molecules coalesce to form incipient soot (blue-circled clusters), which grow further by surface condensation of small free molecules or radicals.**

At high temperature ($T$ = 1650 and 1800 K), acetylene molecules collide more vigorously with each other due to their higher kinetic energy, increasing the probability of bond breakage upon collision. So, linearization and cyclization (Fig. 1c and d: red- and green-circled molecules) occur faster ($t$ = 0.75 ns) than at lower temperatures (Fig. 1a and b). Incipient soot forms rapidly within 2 ns (Fig. 1c and d: blue-circled molecules at $t$ = 2 ns) and grows further by surface condensation for $t \geq 3.75$ ns until the surrounding molecules and radicals are consumed. Therefore, higher temperature leads to faster hydrocarbon cyclization and growth, consistent with diffused back-illumination extinction imaging measurements of pyrolytic decomposition of n-dodecane (Skeen and Yasutomi 2018), revealing a linear increase of the soot formation rate with temperature.

Figure 2 shows the temporal evolution of the molecular weight of the largest molecule or cluster formed



during acetylene pyrolysis at 1350 (black line), 1500 (blue line), 1650 (green line) and 1800 K (red line) for the simulations shown in Figure 1. Initially, only acetylene molecules are present in the simulation domain corresponding to 26 g/mol at $t = 0$ ns at all temperatures. During acetylene pyrolysis, inception takes place slowly by reactive collisions of hydrocarbon molecules. Incipient soot is defined by the formation of soot clusters with

$M_w \geq 202$ g/mol (horizontal line) (Mukut January 2023), corresponding to the molecular weight of pyrene (Dillstrom and Violi 2017), which is one of the most commonly considered seed molecules that initiate soot inception (Frenklach 2002, Mukut January 2023). The onset of surface growth as a function of temperature is shown in Fig. S1. In addition, upon the formation of a molecule with molecular weight of ~202 g/mol, no dissociation of this molecule is observed (as shown exemplarily for $T = 1800$ K, Fig. S2), indicating the transition

from gas to particle phase and the onset of surface growth. This lack of dissociation for clusters larger than ~202 g/mol is observed for all simulations and temperatures. The inception step is slower at low temperature due to low kinetic energy of the colliding reactive species. For example, at 1350 K inception is completed within ~5 ns, i.e., 5 times slower than inception at T $\geq$ 1650 K, that takes place within ~1 ns.

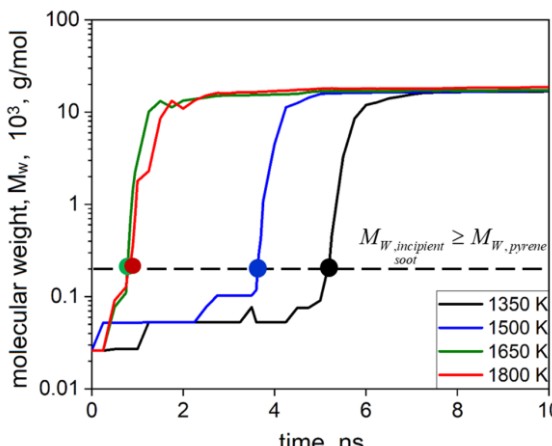

**Figure 2: Temporal evolution of the molecular weight of the largest molecule formed during acetylene pyrolysis at 1350 (black line), 1500 (blue line), 1650 (green line) and 1800 K (red line) for the simulations of Figure 1. The formation of incipient soot is denoted by the formation of soot clusters with $M_w \geq 202$ g/mol (horizontal line), corresponding to the molecular weight of the pyrene molecule.**

Figure 3 shows the temporal evolution of the total number of (a) $C_2H_2$ molecules, as well as compounds containing (b) 1-5 C atoms, $C_1$-$C_5$ (excluding $C_2H_2$), (c) 6-10 C atoms ($C_6$-$C_{10}$) and (d) > 10 C atoms ($C_{10}$) present in the simulation cell, during $C_2H_2$ pyrolysis at 1350 (circles), 1500 (squares), 1650 (diamonds) and 1800 K (stars).



The total number of $C_1$-$C_5$ molecules (including $C_2H_2$) are shown in Figure S3. At $T$ = 1350 K, the consumption of $C_2H_2$ is negligible (< 10 %) up to 4.75 ns. Shortly thereafter, at $t$ = 5.2 ns (Fig. 3a: vertical black line), a sudden drop of 92.2 % in the $C_2H_2$ concentration is observed accompanied by an abrupt increase in the number of other $C_1$-$C_5$ (Fig. 3b: circles) and $C_6$-$C_{10}$ molecules (Fig. 3c: circles). At that time ($t$ = 5.2 ns), a $C_{19}H_{11}$ cluster is formed

5 (Table S1, $T$ = 1350 K: Simulation 1), with molecular weight of 239 g/mol denoting the onset of surface growth (vertical black line), as discussed in Figure 2.

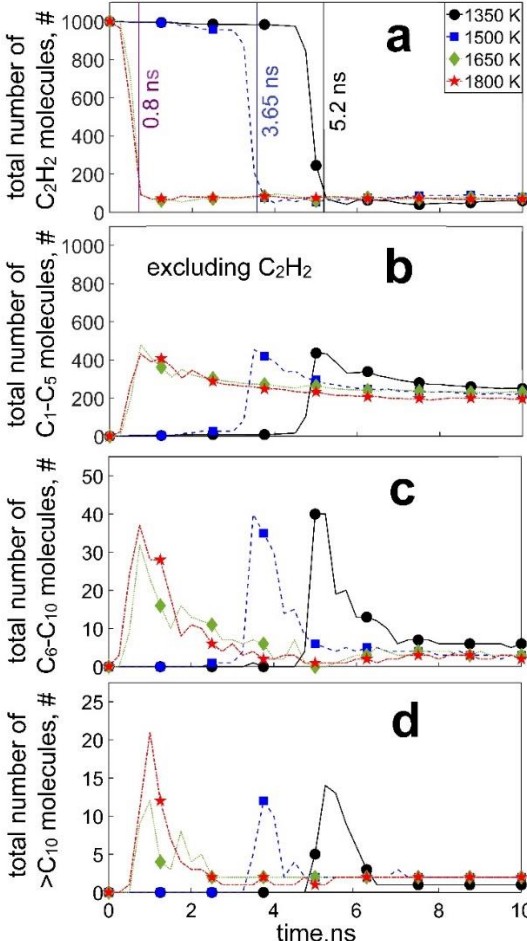

**Figure 3: Temporal evolution of the total number of (a) acetylene ($C_2H_2$), (b) $C_1$-$C_5$ (excluding $C_2H_2$), (c) $C_6$-$C_{10}$ and (d) > $C_{10}$ molecules during acetylene pyrolysis at 1350 (circles), 1500 (squares), 1650 (diamonds)**

10 **and 1800 K (stars).The vertical lines at 0.8 ns for 1650 and 1800 K, at 3.65 ns for 1500 K and at 5.2 ns for 1350 K, indicate the onset of surface growth for the respective temperatures.**



The peak in the number of $C_6$-$C_{10}$ molecules (Fig. 3c) coincides with the increase in the number of $C_1$-$C_5$ molecules other than $C_2H_2$ (Fig. 3b) and the depletion of nearly 90 % of the $C_2H_2$ molecules (Fig. 3a). In fact, narrow peaks appear in the evolution of the number of $C_6$-$C_{10}$, indicating their rapid, practically complete consumption shortly after their formation. These intermediate sized molecules ($C_6$-$C_{10}$) contribute up to 2.5 % of all molecules at the surface growth stage. In contrast, $C_1$-$C_5$ molecules excluding acetylene (Fig. 3b) are consumed at a slow rate, reaching a plateau of about 250 molecules at longer times ($t > 8$ ns), totaling 70-75 % of all molecules, regardless of the temperature. The faster consumption rate of the $C_6$-$C_{10}$ molecules could be attributed to their larger projected area compared to $C_1$-$C_5$ compounds, rendering them more likely to be scavenged by the large incipient soot.

At the onset of surface growth ($t = 5.2$ ns at 1350 K, Fig. 3a: vertical black line), the number concentration of larger molecules, composed of more than 10 C atoms (Fig. 3d: circles) also increases reaching a maximum at $t = 5.25$ ns. As the largest cluster in the simulation domain (e.g., Fig. 1a: $t = 6$ ns) collides with the surrounding species, it grows by scavenging other molecules with large projected area, resulting in a drop in the number of $C_6$-$C_{10}$ (Fig. 3c) and $> C_{10}$ molecules. In contrast to larger molecules consisting of more than 6 C atoms, the concentration of $C_1$-$C_5$ molecules (Fig. 3b: circles) decreases at a much slower rate as the reactive collisions of such molecules with each other and with the soot cluster become more scarce. In fact, for $t \geq 6.5$ ns, only one cluster with $> C_{10}$ is present (Fig. 3d), corresponding to the incipient soot particle (composed of 1134 C atoms).

At higher temperature ($T > 1350$ K), even though soot precursor species form earlier than at 1350 K, a similar growth mechanism is observed. Specifically at $T = 1500$ K, the onset of surface growth takes place at 3.65 ns, corresponding to acetylene reduction of 87.6 % (Fig. 3a: vertical blue line) and to a peak in the number of $C_1$-$C_5$ (Fig. 3b) and $C_6$-$C_{10}$ molecules (Fig. 3c). This delay in the peak concentration of the $C_1$-$C_{10}$ molecules at low temperature is associated with the slower reaction kinetics of acetylene molecules, as indicated by their delayed depletion (Fig. 3a) compared to higher temperature. For $t < 3.65$ ns, small ($< C_{10}$) molecules are mostly formed and only few (less than 10; Fig. 3d) larger molecules are present. At even higher temperature ($T \geq 1650$ K), inception takes place at a much shorter timescale as the onset of surface growth is observed at $t = 0.8$ ns (Fig. 3a: vertical purple line), with a simultaneous increase in $C_{1\text{-}10}$ molecules (Fig. 3b and c), which grow further after 0.2 ns, indicated by the peak of the $> C_{10}$ molecules (Fig. 3d) at $t = 1$ ns.

At sufficiently long times, soot inception and surface condensation are practically completed as the number concentration of all species in the simulation domain is significantly reduced, reaching steady state. At this stage, 16-20 % of all the molecules are $C_2H_2$, 77.5-81.5 % are $C_1$-$C_5$ (excluding $C_2H_2$), and 2.5 % are $C_6$-$C_{10}$ molecules. These MD results have been reproduced with up to four simulations at each temperature using different



initial configurations of acetylene molecules (Figs. S4-S9, simulations $S_1$-$S_4$), confirming the trends shown in Fig. 3. At 1350 K, a large variation is observed in the $C_2H_2$ evolution. The $C_2H_2$ are practically consumed within 3.5 to 8 ns, following a similar trend in all four simulations (Fig. S4), and resulting in qualitatively similar temporal evolutions of the $C_1$-$C_5$ molecules (excluding $C_2H_2$) (Fig. S7), $C_6$-$C_{10}$ (Fig. S8) and > $C_{10}$ molecules (Fig. S9). The

temporal change of the amount $C_1$-$C_5$, $C_6$-$C_{10}$, and > $C_{10}$ species is also shown in terms of molecular wt% in Fig. S10.

Figure 4 shows the temporal evolution of the average C/H ratio of $C_1$-$C_5$ molecules (a) including $C_2H_2$ and (b) excluding $C_2H_2$, (c) $C_6$-$C_{10}$ and (d) > $C_{10}$ molecules for the acetylene pyrolysis simulations of Figure 1, at 1350 (circles), 1500 (squares), 1650 (diamonds) and 1800 K (stars). During inception at $T = 1350$ K ($t \leq 4.5$ ns),

the average C/H ratio of all $C_1$-$C_5$ molecules is 1 (Fig. 4a) as most (> 95 %) of the reactive species is acetylene (Fig. 3a), while the remaining 5 % exhibits a slightly higher C/H ratio of up to 1.3 (Fig. 4b). After the onset of surface growth (i.e., $t > 5.2$ ns at 1350 K), the average C/H ratio of $C_1$-$C_5$ molecules slightly decreases attaining a value of 0.53, including (Fig. 4a) and excluding $C_2H_2$ (Fig. 4b), as most of acetylene has been depleted (Fig. 3a). As larger molecules start to form (Fig. 3c and 3d), the C/H ratio of $C_6$-$C_{10}$ and > $C_{10}$ species increases, reaching a

maximum shortly after surface growth starts ($t \cong$ 5-6 ns). This attainment of the maximum C/H ratio of intermediate sized or large molecules coincides with the maximum number concentration of those compounds throughout soot formation. For $t \geq 6.5$ ns, where steady > $C_{10}$ concentration is reached (Fig. 3d), the average C/H ratio of the > $C_{10}$ molecules is approaching 2 (Fig. 4d). During soot surface growth ($t \geq 5.2$ ns), the C/H ratio of $C_1$-$C_5$ molecules drops, reaching a plateau at 0.57 (Fig. 4a) or at 0.41 without accounting for $C_2H_2$ (Fig. 4b). The

C/H ratio of $C_6$-$C_{10}$ molecules decreases once the incipient soot is formed (Fig. 4c, $t \geq 6.5$ ns) until the attainment of a plateau of about 1. The $C_6$-$C_{10}$ molecules show a higher C/H ratio than that of the $C_1$-$C_5$ molecules during inception. The peak in the C/H ratio of the $C_6$-$C_{10}$ molecules coincides with the cluster formation (i.e., when molecular weight of the incipient soot is ~202 g/mol) indicating that these molecules act as nuclei for the formation of the soot cluster.

Increasing the process temperature leads to faster formation of $C_1$-$C_5$ molecules, which eventually reach the same C/H ratio of 0.41 during soot surface growth (Fig. 4b). This increase in temperature, however, leads to faster attainment of this asymptotic C/H ratio of small (< $C_5$) molecules. Also, higher temperature (> 1650 K) leads to faster formation of $C_2H_4$, $CH_4$, $C_2H_3$ and $C_2H_6$ (please see also Fig. 5b-e), which reduce the average C/H ratio of $C_1$-$C_5$ molecules until a plateau is reached around 3-4 ns (Fig. 4b). Since no stable > $C_5$ molecules are

formed before the incipient soot formation (Fig. 1, 2, 3c and 3d), the C/H ratio for $C_6$-$C_{10}$ molecules is mainly observed post-inceptions for all temperatures and, only when > $C_5$ molecules are formed, the C/H ratio slightly

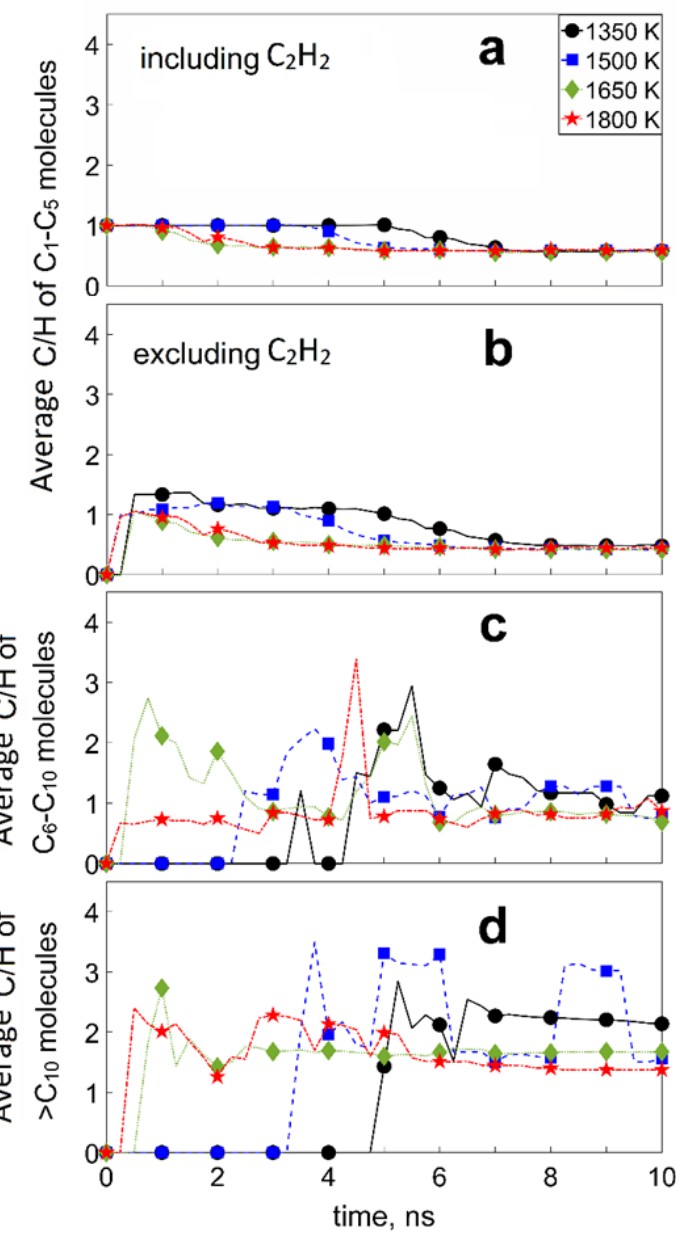

**Figure 4: Temporal evolution of the average C/H ratio of $C_1$-$C_5$ molecules (a) including $C_2H_2$ and (b) excluding $C_2H_2$, (c) $C_6$-$C_{10}$ and (d) > $C_{10}$ molecules for the acetylene pyrolysis simulations of Fig. 1, at 1350 (circles), 1500 (squares), 1650 (diamonds) and 1800 K (stars).**





fluctuates as the cluster grows (Fig. 4c and 4d). Additionally, the > $C_{10}$ (Fig. 4d) and $C_6$-$C_{10}$ (Fig. 4c) molecules are more carbonized than their smaller counterparts ($C_1$-$C_5$ molecules), indicated by their higher C/H ratio. The C/H ratio of the incipient soot during acetylene pyrolysis (Fig. 4d) is lower at higher temperature (≥1650 K) due to dehydrogenation, consistent with experiments with other fuel flames (e.g., methane (Alfè et al. 2009, Alfè et al. 2010, Russo, Tregrossi and Ciajolo 2015, Russo et al. 2013) and ethylene (Alfè et al. 2009, Russo et al. 2015, Russo et al. 2013)). The C/H evolutions have been reproduced for up to four MD simulations with various initial configurations (Figs. S11-S14), confirming the trends shown in Fig. 4.

Figure 5 shows the temporal evolution of the number of (a) $C_2$ and C, (b) $C_2H_4$ and $C_4H_2$, (c) $CH_4$ and $CH_3$, (d) $C_2H_3$ and $C_2H$, (e) $C_2H_6$ and $C_2H_5$, corresponding to the 10 most abundant organic species and (f) $H_2$ and H formed by acetylene pyrolysis during inception and surface growth at 1350 (black lines), 1500 (blue lines), 1650 (green lines) and 1800 K (red lines). The concentration of all molecules increases while $C_2H_2$ is rapidly consumed (Fig. 3a) and, at the onset of surface growth, $C_2H_3$ exhibits the highest concentration (Fig. 5d, solid line), suggesting that the intermediates formed by reactive collisions of acetylene are eventually converted to vinyl radicals, as proposed in the hydrogen abstraction vinyl acetylene addition (HAVA) mechanism (Shukla 2012).

For example, at $T$ = 1350 K, the $C_2H_3$ (Fig. 5d) and $C_2H_4$ exhibit a peak (Fig. 5b) at $t$ = 5.25 ns, which corresponds to the timestep right after the onset of the surface growth (Fig. 3). There ($t$ = 5.25 ns), only a small amount of $C_2H$ (Fig. 5d), $C_4H_2$ (Fig. 5b), $C_2$, C (Fig. 5a), and $H_2$ (Fig. 5f) is produced, followed by an increase in $CH_3$ and $CH_4$ (Fig. 5c) at $t$ ≥ 5.5 ns, indicating that the primary products during acetylene pyrolysis (such as $C_2H_3$, $C_2H_4$) also contribute to the formation of hydrocarbon molecules containing > 5 C. The amount of $C_2H_4$ (Fig. 5b: solid lines), $CH_4$ (Fig. 5c: solid lines), $C_2H_6$ (Fig. 5e: solid lines), and $H_2$ (Fig. 5f: solid lines) hardly changes at longer times, after $C_2H_2$ has practically been depleted, indicating that these species do not contribute to the growth of the soot cluster. The abrupt decrease in $C_2H_4$ (Fig. 5b: solid lines), $C_2H_3$ (Fig. 5d: solid lines) and $C_2H$ concentration (Fig. 5b: broken lines) also hints that these species contribute towards the growth of incipient soot. The process temperature hardly affects the concentration of the organic species, but higher temperatures lead to faster reactions as discussed in Fig. 3. It is worth noting that at low temperature ($T$ = 1350-1650 K), dehydrogenation takes place only right after the onset of surface growth and the $H_2$ concentration remains constant at longer times. At $T$ = 1800 K, however, even though dehydrogenation starts at the onset of surface growth, coinciding with the formation of $C_6$-$C_{10}$ molecules (Fig. 3c), it continues throughout surface growth of the incipient soot as $H_2$ increases for $t$ ≥ 4 ns (Fig. 5f: $T$ = 1800 K, solid line). The large $H_2$ concentrations generated by pyrolysis at high T is consistent with the $H_2$ synthesis in plasma reactors for carbon black production (Fulcheri and Schwob 1995).



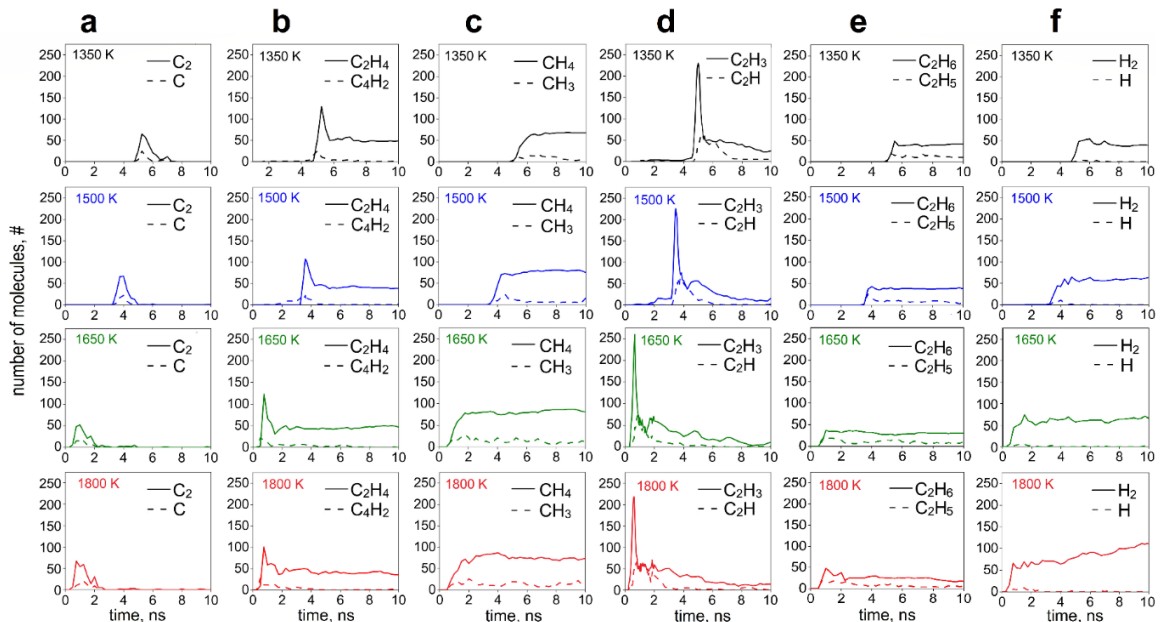

**Figure 5: Temporal evolution of the number of (a) C₂ and C, (b) C₂H₄ and C₄H₂, (c) CH₄ and CH₃, (d) C₂H₃ and C₂H, (e) C₂H₆ and C₂H₅ and (f) H₂ and H corresponding to the most abundant species formed by acetylene pyrolysis during inception ($t = 0 – 10$ ns) at 1350 (black lines), 1500 (blue lines), 1650 (green lines) and 1800 K (red lines).**

Figure 6 shows the temporal evolution of the cumulative number of (a) 3-, (b) 5-, and (c) 6-member C rings across all molecules during acetylene pyrolysis at 1350 (circles), 1500 (squares), 1650 (diamonds) and 1800 K (stars). The appearance of the 3-member rings occurs at $t = 3.75$ ns at 1350 K after $C_2H_2$ has practically been depleted (Fig. 3a) and coincides with the formation of $C_6$-$C_{10}$ molecules (Fig. 3c), indicating the presence of 3-member rings in molecules consisting of 6-10 C atoms. The total number of 3-member rings decreases when small molecules are consumed and the incipient soot grows, revealing that 3-member rings are thermodynamically less stable (Kim and Ihee 2012, Fantuzzi et al. 2013) compared to the acyclic small molecules ($C_1$-$C_5$) and dissociate when surface growth prevails. The 5- and 6-member rings are formed simultaneously during pyrolysis. Early on, before the onset of surface growth, no 5- and 6-member rings are observed, indicating that these rings exist mainly in the incipient soot. For example, at $T = 1350$ K (Fig. 6c), 6-member rings start to form at $t = 6$ ns after more than 60 % of the $C_6$-$C_{10}$ molecules have been depleted (Fig. 3c: $t = 6$ ns). At longer times, where incipient soot



cluster growth steadies, the concentration of 5- and 6-member rings reaches a plateau, corroborating the fact that these rings originate from the formation of the incipient soot.

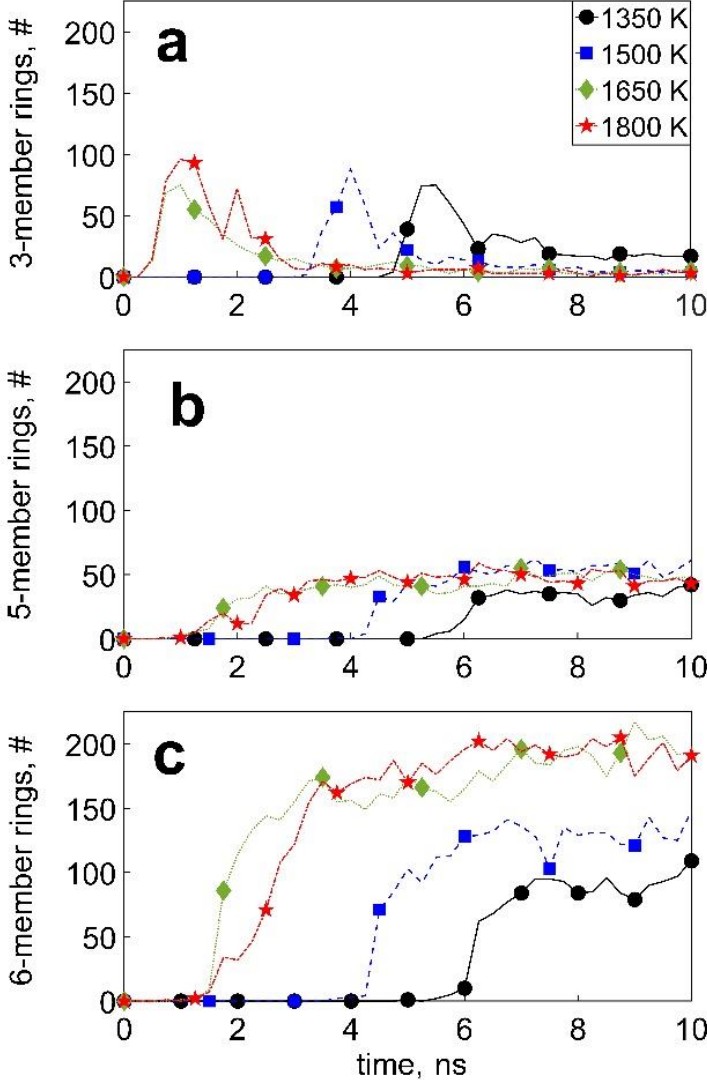

**Figure 6: Temporal evolution of the number of (a) 3-, (b) 5-, and (c) 6-member C rings during acetylene pyrolysis at 1350 (circles), 1500 (squares), 1650 (diamonds) and 1800 K (stars). The appearance of the 3-member rings occurs after $C_2H_2$ has been depleted. The total number of 3-member rings decreases when small molecules are consumed, and the incipient soot grows while the number of 5- and 6-member rings increases.**



The formation of 5- and 6-member rings is tracked exemplarily in Fig. S15 for different molecules at $T$ = 1500 K. The initial small hydrocarbons polymerize by reactive collisions, forming extended aliphatic chains consisting of more than 6 C atoms. These large aliphatic chains undergo molecular rearrangement and cyclization, resulting in the formation of 5-member and 6-member aromatic rings (e.g., Figure S15a: $t$ = 7.55 and 7.75 ns).

These rings dynamically form and may dissociate; for instance, the rings highlighted in Figure S15a persist for approximately 0.25 ns before dissociating due to collisions with other molecules. Concurrently, other large aliphatic molecules coalesce with existing rings, stabilizing the incipient soot nanoparticle (Figure S15b: $t$ = 9.2). This leads to further ring formation through molecular rearrangement, rather than through PAH dimerization, thereby underscoring the complex and transient nature of soot precursor dynamics at high temperature.

At the onset of surface growth, the 5-member aromatic rings are approximately 13 % for 1350 K and 15 % for 1500, 1650 and 1800 K of the total number (aromatic and aliphatic) of the 5-member rings formed (Fig. S16a). At longer times, the fraction of 5-member aromatic rings drops down to 3 % for 1350 K and 2 % for 1500, 1650 and 1800 K (Fig. S16a), while the fraction of 6-member aromatic rings is 6 % for 1350 and 1650 K and 5 % for 1500 and 1800 K (Fig. S16b). As time proceeds, the fraction of 6-member aromatic rings rises to 12 and 10 

% for 1350 and 1500 K and 11 % for both 1650 and 1800 K, consistent with recent MD results (Han et al. 2017). In addition, the total number of 6-member rings increases with increasing temperature (Fig. 6c) and cyclization within the incipient soot takes place.

This progression of ring formation is reflected in the size (Fig. S17) and molecular weight distributions (Fig. S18) of the population of all species during soot inception. At 1350 K for t ≤ 3.75 ns and at 1500 K for $t \leq 2$ 

ns, more than 95 % of molecules are acetylene, so there is hardly any change in the molecular weight distribution and only a few of them have molecular weights up to 200 g/mol, in line with Figures 1 and 3a. At longer times, i.e., at 1350 K and $t \geq 5$ ns (Fig. S17a and S18a) and at 1500 K and $t \geq 3.75$ ns (Fig. S17b and S18b), the distributions shift to larger cluster sizes and molecular weights, with the formation of molecules, radicals, and even a nascent soot cluster having molecular weight between 200 and 1,000 g/mol. The peak corresponding to the 

largest cluster shifts towards 10,000 g/mol for $t \geq 6$ ns at 1350 K (Fig. S18a) and for $t \geq 5$ ns at 1500 K (Fig. S18b). Likewise, at 0.75 ns for 1650 (Fig. S18c) and 1800 K (Fig. S18d), the molecular weight distribution of reactive components spans between 100 to 1000 g/mol, indicating that nucleation has practically stopped, and nascent soot clusters have been formed. At 6 ns and beyond, the growth of the soot cluster stops (with molecular weight peaks being observed around 10,000 g/mol) due to depletion of the reactive species.



### 3.2 Chemical structure of soot precursors

Tables 2-5 show a breakdown of the detailed chemical structure of the molecules generated during acetylene pyrolysis at 1500 K for 0.75 and 2 (Table 2), 3.75 (Table 3), 5 and 6 (Table 4), and 8 ns (Table 5) for the simulation shown in Fig. 1b. For clarity, the incipient soot is omitted at 5, 6 and 8 ns.

Early on ($t \leq 2$ ns; Table 2), only small molecules consisting of up to 4 C atoms are observed. Among those, acetylene is the most abundant (Fig. 3a), while $H_2$, $C_2H$ (cyclic and linear), $C_2H_3$, $C_2H_4$, $C_4H_2$ (as shown in Fig. 5), $C_4H_3$ and $C_4H_4$ are also present at low concentrations. Shortly after the onset of surface growth, at $t = 3.75$ ns (Tables 3 and S2), two ($C_{84}H_{39}$ and $C_{87}H_{40}$) soot nuclei appear (Table S2). Parallel to this, both cyclic and linear molecules with more than 6 C atoms emerge with most of the cyclic structures consisting of 3-member rings

(highlighted in yellow). Some of these 3-member rings reported in Table 3 correspond to known monocyclic molecules, including methylene cyclopropane (Zádor et al. 2017) (molecule A), cyclopropyne (Saxe and Schaefer 1980) (molecule B), and cyclopropane (Li, Zhang and Shi 2020) (molecule C), and bicyclic molecules, including bicyclo[1.1.0]butane (Fawcett 2020) (molecule D), spiropentadiene (Billups and Haley 1991) (molecule E) and cyclopropylidenecyclopropane (Güney et al. 2013) (molecule F). $C_2H$ cyclic molecules (highlighted in pink)

having three-center two-electron configuration (Lammertsma and Ohwada 1996) are observed at all six timesteps indicating they are relatively stable in nature. Cyclobutane (highlighted in green) is detected only at 3.75 ns (Table 3) indicating its thermodynamic instability, consistent with Zador et al. (Zádor et al. 2017). Aliphatic compounds such as propane (Iijima 1972), propene (Lide and Christensen 1961), butane (Bradford, Fitzwater and Bartell 1977), butene (Lu, Li and Lu 2017) and other alkanes, alkenes and alkynes are also observed at 3.75 ns or later

(Tables 3-5). The 5- (blue shaded rings) and 6-member (red-shaded rings) rings typically manifest in $> C_{10}$ molecules (Tables 3 and S2). The small molecules ($H_2$, $C_2H$, $C_2H_2$, $C_2H_3$, $C_2H_4$, $C_4H_2$, $C_4H_3$ and $C_4H_4$) that have formed at earlier stages, persist at 3.75, 5 and 6 ns (Tables 3 and 4), albeit with a notable reduction in their number concentration.

Due to their thermodynamic instability, the 3-member rings formed early on (e.g., Table 3: $t = 3.75$ ns),

dissociate almost immediately, as indicated by the drop in their number at 5, 6 and 8 ns, while molecules other than the incipient soot possess only a few 5- or 6-member rings during these timesteps (Tables 4 and 5). However, the total number of 5- or 6-member rings in the entire population of soot precursor molecules is much higher (approximately 50 5-member rings, Fig. 6b; and 100-150 6-member rings, Fig. 6c), suggesting that almost all of them belong to the incipient soot.

At the initial stages of soot inception ($t = 0.75$ and 2 ns), linear and branched aliphatic hydrocarbons are formed along with unsaturated carbon chains and radical sites. One of the most abundant radicals formed is ethenyl

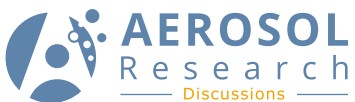

($C_2H_3$, Fig. 5d). Due to its rapid formation and almost immediate consumption, ethenyl is crucial for further growth of hydrocarbon structures and soot inception, consistent with Wang et al. (Wang et al. 2021). By adding to unsaturated carbon bonds, ethenyl can facilitate chain propagation and cross-linking for cyclization, leading to the formation of new rings, as evidenced by molecule $C_{26}H_{15}$ at 3.25 ns (Table 3) and molecule $C_{19}H_{32}$ at 8 ns (Table 5).

The present results highlight the relevance of chemical nucleation in soot formation at intermediate temperatures (1350 –1800 K), emphasizing a substantial role for chemical pathways even in the absence of observable physical PAH dimerization. This is in stark contrast to ReaxFF simulations that start with PAH as monomers (Mao et al. 2017), indicating lack of nucleation at temperatures ranging from 1200 – 2400 K, despite soot yield peaks within this temperature range (Frenklach et al. 1983).

**Table 2. Chemical structures of all molecules formed during acetylene pyrolysis at $t$ = 0.75 and 2 ns at 1500 K. Molecules larger than $C_4$ are not formed within the first 2 ns of acetylene pyrolysis at this temperature.**

| $t$, ns | $C_1 – C_5$ | $C_6 – C_{10}$ | > $C_{10}$ |
|---|---|---|---|
| 0.75 | | - | - |
| 2 | | - | - |

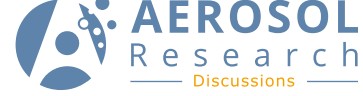

The detailed chemical structure of molecules formed during acetylene pyrolysis is also shown for 1350, 1650 and 1800 K at 0.75, 2, 3.75, 5, 6 and 8 ns (Supporting Information: Tables S3-S15). For $t \leq 3.75$ ns, formation of vinyl molecule ($C_2H_3$) and cyclic $C_2H$ molecule is observed along with small aliphatic molecules (< 5 C atoms) at 1350 K (Supporting Information: Table S3). At $t = 5$ ns (Supporting Information: Table S4), 3-member rings and molecules containing > 5 C atoms are formed, coinciding with $C_2H_2$ consumption (Fig. 3a). At 1350 K (Supporting Information: Tables S3-S6), hardly any 5- and 6-member rings are found in the reported chemical structures (at 5 ns, there is a molecule with a 6-member ring in the $C_6$-$C_{10}$ range, Supporting Information: Table S4), which exclude the incipient soot, indicating that most of these rings are formed within the incipient soot nanoparticle. However, smaller benzene derivatives are observed in the reaction pathway at 1650 (Tables S10-S11) and 1800 K (Tables S14-S15).

At both 1650 (Tables S7-S11) and 1800 K (Tables S12-S15), mostly 3-member rings are formed up to 6 ns along with the incipient soot (excluded in the tables). At $T = 1650$ K, a benzene derivative forms at 5 ns, while naphthalene derivatives form at 6 and 8 ns (Supporting Information: Table S11). However, only a few small and intermediate molecules with 5- and 6-member rings exist at $t = 6$ and 8 ns at 1800 K (Supporting Information: Table S15), further indicating that most of these rings belong to the large incipient soot nanoparticle. At all temperatures employed here (1350-1800 K), large molecules with more than 10 C atoms are composed of a few (5- and 6-member rings) decorated with long branches (Tables 3-5, S2, S3-S6, S7-S11 and S12-S15). Even though these structures are different from the majority of those observed experimentally (Lieske et al. 2023, Commodo et al. 2019, Jacobson et al. 2020, Martin et al. 2021), which are composed of aromatic islands with peripheral methyl groups, long alkyl chains have also been observed (Schulz et al. 2019) on par with the present simulations. It should also be noted that at temperatures below 1800 K, which are relevant to soot formation in flames, the ReaxFF-predicted cluster structures are significantly different from those formed by acetylene combustion at higher temperature (2700 K) (Wang et al. 2022a), by methane and ethylene combustion at 3000 K (Wang et al. 2022b), or by dimerization of PAHs (Zhao et al. 2020),where large carbonaceous clusters are composed mainly of rings and contain only a small fraction of branches.





**Table 3: Chemical structures of all molecules, composed of up to 25 carbon atoms, formed during acetylene pyrolysis at 3.75 ns at 1500 K. Two larger nuclei $C_{84}H_{39}$ and $C_{87}H_{40}$ are also present (shown in Table S2).**



**Table 4: List of all molecules, aliphatic, cyclic, or aromatic, except the incipient soot, formed during acetylene pyrolysis at 5 (top row) and 6 ns (bottom row) at 1500 K, including their chemical structures. The excluded incipient soot consists of 1266 and 1307 C atoms at 5 and 6 ns, respectively.**

| *t*, ns | $C_1 - C_5$ | $C_6 - C_{10}$ | $C_{10} - C_{25}$ |
|---|---|---|---|
| 5 | | | |
| 6 | | | |



**Table 5: List of all molecules, aliphatic, cyclic, or aromatic molecules, except the incipient soot (consisting of 1335 C atoms) formed during acetylene pyrolysis at 8 ns at 1500 K, including their chemical structures.**

| $C_1 - C_5$ | $C_6 - C_{10}$ | $C_{10} - C_{25}$ |
|---|---|---|



(C_{19}H_{32})

## 4 Conclusions

The inception and early stages of soot surface growth by acetylene pyrolysis are investigated using reactive molecular dynamics at 1350-1800 K. These simulations do not assume any specific soot precursors, allowing for a broader and more accurate exploration of the pathways involved in soot nucleation. Increasing the temperature leads to faster formation of the incipient soot taking place through linearization, cyclization, and subsequent surface condensation of radicals on the incipient soot. During linearization and cyclization, small molecules consisting of less than 6 C atoms, such as $C_2H_3$, $C_2H_4$, $C_2H_6$, $CH_4$ and $C_2$ are formed at all temperatures. These small $C_3$-$C_6$ molecules are mainly aliphatic chains or 3-member rings. Increasing the process temperature leads to faster depletion of $C_2H_2$ molecules and faster formation of these compounds. The growth of small $C_1$-$C_5$ molecules

can be attributed to reactive collisions, which eventually lead to the formation of larger aliphatic compounds consisting of 6-10 C atoms. At the initial stages of inception and prior to the formation of the incipient soot, 3-member rings are formed associated with the formation of compounds with less than 10 C atoms. Once the incipient soot is formed, the number of $C_1$-$C_{10}$ compounds and the number of 3-member rings drops, while the

number of 5- and 6-member rings increase, indicating that the formation of larger rings is associated with the formation and growth of the incipient soot. The cyclic structures are mainly observed within the incipient soot which is supported by the information of the detailed chemical structures of the molecules observed at different timesteps. Most of these cyclic structures consist of a few rings interconnected with aliphatic side chains. By providing a comprehensive list of all molecular precursors—rather than focusing solely on the most abundant

ones—this study offers a more complete view of the chemical complexity involved in soot nucleation. Tracking the pathway of formation of these species could reveal new detailed chemical routes that occur during soot nucleation, which might have not been extensively considered in existing kinetic models, thereby expanding the current understanding of soot formation pathways.

**Conflict of Interest**

E.G. is a member of the editorial board of Aerosol Research. The remaining authors declare no conflict of interest.

**Acknowledgments**

The research benefited from computational resources provided through the NCMAS, supported by the Australian

Government, and The University of Melbourne's Research Computing Services and the Petascale Campus Initiative. A.G. acknowledges Felix Schmalz for his assistance with ChemTraYzer software. K.M.M. and S.R. acknowledge funding support from the National Science Foundation as some of this material is based upon work supported by the National Science Foundation under Grant No. 2144290.

**Supporting Information**

Figures showing the onset of incipient soot, temporal evolution of acetylene and other hydrocarbons, composition (C/H ratio), 5- and 6-member aromatic rings, size distribution of molecules as a function of volume-equivalent diameter and molecular weight distribution of molecules at 1350-1800 K as well as tables showing the incipient soot for all simulations and the chemical structures of the hydrocarbons at various timesteps at 1350, 1650 and

1800 K during acetylene pyrolysis.



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
