# Peer review of "Investigation of soot precursor molecules during inception by acetylene pyrolysis using reactive molecular dynamics"

_Aerosol Research, 2024_

## Author Comment (AC1)

**Response ar-2024-34:** "**Investigation of soot precursor molecules during inception by acetylene pyrolysis using reactive molecular dynamics**" by Ganguly et al. The Editor's and Reviewers' comments are shown in *italics*. The responses (blue) and changes in the revised manuscript (red), are given below. Corrections made on this new revision were highlighted in red.

**Reviewer 1**

*The current work reports on ReaxFF simulations of acetylene from 1350-1800 K for large systems (1000 molecules) for long time spans (10 ns). The work reports statistics of molecules formed over time with emphasis on the formation of incipient soot, i.e. molecule growing to large species with a molecular weight over roughly 200 up to about 1000 in this work. The analysis of the ReaxFF simulations is quite detailed and some kinds of molecules are found that have not yet been reported in similar studies at different conditions (e.g. species with small rings and long aliphatic chains). There is some comparison to and new insight compared to previous ReaxFF simulations of soot formation.*

*A1. However, there is little comparison to experiments results (some are mentioned and some disagreement is found, but not discussed in much detail (p. 19))…*

A1. We thank the Reviewer for this Comment. Indeed, comparison of ReaxFF simulations with experiments is challenging due to lack of experimental data in the sub-nm scale. We have provided a comparison of the C/H ratio of the MD-generated soot precursors with previously reported AFM measurements in Fig. S19 of the original manuscript.

To address the Reviewer's Comment, we have added the following discussion in p. 14, lines 8-23 of the revised manuscript: "The C/H ratio of the MD-obtained non-cyclic (Fig. S19: diamonds) and cyclic soot precursors (Fig. S19: circles) consisting of 11-70 C atoms formed throughout the nucleation and surface growth stages ($t$ = 0-10 ns) is compared to that of soot precursors obtained during ethylene combustion by high-resolution atomic force microscopy (AFM) measurements (Commodo et al. 2019, Lieske et al. 2023) (Fig. S19: filled symbols) at 1350 – 1800 K. Most of the soot precursor molecules composed of up to 25 C atoms exhibit a C/H ratio in the range of 1-2, consistent with the C/H ratio of the molecules observed by AFM (Commodo et al. 2019, Lieske et al. 2023), which range between 1 to 2.5. Larger MD-obtained clusters attain a C/H ratio of 1-1.5, contrary to experiments revealing higher C/H ratios of ~2. This can be attributed to the larger fraction of aromatic rings observed in the soot clusters sampled from premixed ethylene ($C_2H_4$)/air flames (Commodo et al. 2019, Schulz et al. 2019), along with the fact that side-aliphatic chains had been excluded from the calculation of the C/H ratio of these AFM-obtained soot clusters (Commodo et al. 2019). This difference between AFM measurements (Commodo et al. 2019, Lieske et al. 2023) and MD simulations could also arise from the presence of $O_2$ in experiments that might have led to the formation of different soot precursors than those obtained here by pyrolysis alone. Aliphatic moieties, however, have been observed both in AFM mainly in the form of methyl groups or larger alkyl chains (Schulz et al. 2019). Additionally, the characteristics of soot nanoparticles obtained at the end of these simulations have been rigorously validated in Mukut et al. (Mukut et al. 2024), further confirming the validity of the present simulations.".

*A2a. …and none to soot modelling based on kinetic mechanisms / models, though many exist for acetylene pyrolysis ("acetylene soot formation models" in google scholar yields many results with promising titles for work on and with such models). I think such a comparison would strengthen the work considerably,…*

A2a. We thank the Reviewer for raising this question. Even though many reaction kinetics mechanisms exist in the literature for soot formation by acetylene, they primarily rely on dimerization reactions and PAH formation with subsequent physical dimerization of PAHs that leads to soot formation. These kinetic models provide an Arrhenius approximation for some reaction pathways and rely on some fundamental calculations and a lot of experimental observations.

In the present simulations, most of the investigated soot precursor molecules are composed of cyclic structures decorated will aliphatic branches (e.g., Tables 1-5), in contrast to existing kinetic models which capitalize on formation of peri-condensed PAHs. It should be noted, however, that even though not present at all timesteps, aromatic species formation is observed. For example, in one of the acetylene pyrolysis simulations at 1500 K, a benzene ring formation is observed at 7.55 ns, as demonstrated in Figure S15b (bottom) and Figure R1 below:

[Figure]

**Figure R1.** Snapshots of molecules leading to formation of a benzene molecule at 1500 K.

This is consistent with kinetic models suggesting that benzene, a major product of acetylene pyrolysis, is formed by addition of $C_2H_2$ on vinylacetylene, as discussed in reaction R12 of Saggese et al. (Saggese et al. 2014).

To further address the Reviewer's Comment, the following discussion has been added in p. 18 lines 9-11 of the revised manuscript: "The present ReaxFF simulations can capture the formation of benzene, a major intermediate of soot formation (Fig. S15b, bottom), consistent with reaction mechanisms of acetylene pyrolysis (Saggese et al. 2014)".

*A2b. ...together with a quantum mechanical validation of new pathways or at least a note of caution that the results pertain to the ReaxFF model used, and not necessarily to reality.*

A2b. The scope of our work focuses of the characterization of possible soot precursor molecules rather than exploring possible reaction pathways. For the latter, a more detailed and automated comparison of the ReaxFF-based reaction network to kinetic models would be required with reaction path identification analysis. To the best of our knowledge, such analysis has been made possible only for short simulation times as it is computationally extremely demanding since it requires recording geometry and atomic connectivity data every 1 fs (Schmalz et al. 2024), to capture C–H bond vibrations. In the present simulations this information is recorded every 0.25 ns, as mentioned in the first sentence of Section 2.2: "The chemical structure of all species consisting of up to 87 C atoms formed during pyrolysis and their number concentration is obtained as a function of time and is recorded every 0.25 ns." rendering reaction pathway identification impossible.

It should also be noted that reactive MD simulations provide detailed elementary reactions, while kinetic reaction mechanisms are typically based on lumped steps of elementary reactions. As such, the present ReaxFF simulations can serve as a tool to propose new possible additions in literature chemical reaction pathways. This has been demonstrated by Schmalz et al. (Schmalz et al. 2024) for n-heptane and iso-octane pyrolysis by ReaxFF, revealing that >90% of ReaxFF-obtained reactions were not considered in kinetic models (Langer, Mao and Pitsch 2023).

To clarify this, the following discussion has been added in the 3rd paragraph of the Introduction of the revised manuscript: "…based on ReaxFF force field (Castro-Marcano et al. 2012) provide insight into the reaction kinetics (Schmalz et al. 2024) and dynamic formation of soot (Goudeli 2019). Schmalz et al. (Schmalz et al. 2024) identified the reaction pathways to the formation of benzene, predicted by ReaxFF pyrolysis of *n*-heptane and *iso*-octane, revealing that >90% of ReaxFF-obtained reactions were not considered in kinetic models (Langer et al. 2023). Such reaction path identification analysis, however, has been limited only to short simulation times and 100 fuel molecules, as even such small pyrolysis systems yield a complex network of more than 10,000 reactions. ReaxFF simulations have been used more commonly to investigate soot inception (Mao, van Duin and Luo 2017, Yuan et al. 2019, Han et al. 2017) and growth (Yuan et al. 2019) using PAHs as the starting nucleating species. These simulations…".

As pointed out by the Reviewer and as discussed in Schmalz et al. (Schmalz et al. 2024), a system-specific verification of the ReaxFF-derived reactions by *ab initio* quantum mechanics calculations, such as density functional theory, is necessary. To address this, the following sentence was added in the end of the Conclusions of the revised manuscript: "However, verification of ReaxFF-derived reactions with *ab initio* quantum mechanics calculations, such as density functional theory is essential to ensure the importance of these reactions in soot formation kinetics.".

*Specific comments:*

*1. P 5: Please make the scripts used for analysis of the MD data available or add more information on the algorithms used.*

1. Post-analysis of the MD data was carried out using the open-source Chemical Trajectory Analyzer (ChemTraYzer) analysis tool (Döntgen et al. 2018) tailored for post-processing reactive Molecular Dynamics. ChemTraYzer utilizes bond order information and atom trajectories generated by ReaxFF simulations. The individual molecules emerging at each timestep are distinguished based on the atom connectivity data available in the bond order files generated by ReaxFF simulations, and their detailed chemical structure is visualized

based on the ReaxFF-obtained bond order information based on their interatomic distances (Chenoweth, van Duin and Goddard 2008). In ChemTraYzer, the bond orders are rounded to increments of 0.5 (i.e., 0.5, 1, 1.5, etc.) with those below 0.5 being disregarded (Krep et al. 2022). If an atom's total rounded bond orders exceed its valency, no further bonds are assigned. The accepted bonds are then processed through Open Babel (O'Boyle et al. 2011) to obtain the simplified molecular-input line-entry system (SMILES) code.

To clarify this, we have modified Section 2.2 of the revised manuscript as follows: "The chemical structure of all species consisting of up to 87 C atoms formed during pyrolysis and their number concentration is obtained as a function of time and is recorded every 0.25 ns. The detailed structure of each individual molecule is visualized using the Chemical Trajectory Analyzer (ChemTraYzer) analysis tool (Döntgen et al. 2018, Döntgen et al. 2015), which utilizes the bond order information and the atom coordinates generated by ReaxFF simulations. The individual molecules emerging at each timestep are distinguished based on the atom connectivity data available in the bond order files generated by ReaxFF simulations, and their detailed chemical structure is visualized based on the ReaxFF-obtained bond order information and their interatomic distances (Chenoweth et al. 2008). In ChemTraYzer, the bond orders are rounded to increments of 0.5 (i.e., 0.5, 1, 1.5, etc.) with those below 0.5 being disregarded, as discussed in Krep et al. (Krep et al. 2022). The bond information is converted to simplified molecular-input line-entry system (SMILES) codes through Open Babel (O'Boyle et al. 2011) representing the detailed chemical structures of each of the molecules, which are visualized with MolView (Bergwerf 2015). The snapshots of the entire simulated system were visualized using VMD (Humphrey, Dalke and Schulten 1996).".

In addition, the algorithm for ring analysis is discussed in a new Section 2.3 in the revised manuscript:
"**2.3 Analysis of cyclic structures**
The total number of 3-, 5-, and 6-member rings formed in the system is quantified at different timesteps during the pyrolysis simulations. A connectivity matrix is constructed utilizing the coordinates and bond order of all atoms and their neighbors. When 3, 5, or 6 atoms are connected in series in a closed loop they correspond to 3-, 5- and 6-member rings, respectively, and are distinguished from non-cyclic structures.

Once the cyclic structures have been identified, the bonds of each of their constituent pairs are categorized as single C-C bonds if the bond order is < 1.33 (Emri and Lente 2004), double C-C bonds if the bond order values range between 1.33-2.85 (Emri and Lente 2004, Hermann and Frenking 2016), and triple C-C bonds if the bond order is at least 2.86, corresponding to the triple C-C bond in $C_2H_2$ (Hermann and Frenking 2016). A bond order greater or equal to 1.33 (graphite) (Emri and Lente 2004) indicates the presence of a double bond for aromatic compounds and bond order of ~1.92 (ethylene) indicates double bonds in aliphatic compounds (Hermann and Frenking 2016). The aromaticity of each of the ring structures is assessed by counting the electrons in p orbitals that are involved in double bonds or lone pairs and applying Hückel's rule (Solà 2022). Due to shared electron distribution and fractional bonds predicted by ReaxFF, along with the assignment of bond orders in increments of 0.5 by ChemTraYzer, some molecules might be visualized as having C atoms with more than four bonds, which have been excluded from the reported results.".

*2. P 8: "Incipient soot is defined by the formation ..." providing this definition earlier (before talking about incipient soot) may make it easier to follow the introduction.*

2. We thank the Reviewer for this suggestion. In the revised manuscript, we moved the definition of incipient soot in the 1st paragraph of Section 2.1: "…used in this study. Here, incipient soot is referred to soot clusters with $M_w \geq 202$ g/mol (Mukut et al. 2023), corresponding to the molecular weight of pyrene (Dillstrom and Violi 2017), which is one of the most commonly considered seed molecules that initiate soot inception (Frenklach 2002, Mukut et al. 2023). Bond breakage and new…" and we rephrased the corresponding clause in p. 8 of the "Results and Discussion" Section: "…hydrocarbon molecules. The formation of incipient soot, defined as soot clusters with molecular weight equal to or greater than that of pyrene (Dillstrom and Violi 2017), is denoted by the horizontal line in Fig. 2, corresponding to $M_w = 202$ g/mol. The onset of surface growth…".

*3. P 10: "The faster consumption rate of the C6-C10 molecules could be attributed to their larger projected area compared to C1-C5 compounds, rendering them more likely to be scavenged by the large incipient soot." (And a similar statement in the following paragraph) This seems very speculative to me. Considering the much higher number of C1-5 molecules and the moderate size difference, I would expect the opposite (more collisions). Please check by e. g. calculating collision frequencies with a simple kinetic gas theory model if this is the correct reason.*

3. The collision frequency based on the kinetic theory of gases can be estimated by:

$$\beta = \pi \left(d_1 + d_2\right)^2 \sqrt{\frac{k_B T}{2\pi}\left(\frac{1}{m_1} + \frac{1}{m_2}\right)}$$

(R1)

where $k_B$ is the Boltzmann constant, $T$ is the temperature, and $d_1$ & $d_2$ are the diameters of colliding species 1 & 2, corresponding to masses $m_1$ & $m_2$.

To compare the consumption rates of $C_{1-5}$ and $C_{6-10}$ molecules, we calculated the collision frequencies of acetylene ($C_2H_2$), with kinetic diameter of 0.33 nm and a molecular mass of 26.038 g/mol, and naphthalene ($C_{10}H_8$) with kinetic diameter of 0.62 nm and a molecular mass of 128.17 g/mol. Equation R1 yields $\beta_{acetylene} = 3.58 \cdot 10^{-16}$ m$^3$/s and $\beta_{naphthalene} = 5.7 \cdot 10^{-16}$ m$^3$/s for acetylene and naphthalene, respectively. This result indicates that the largest molecules are indeed consumed faster than smaller ones, assuming that every collision is reactive leading to successful clustering.

To further support this statement, we have added the following sentence in the end of the first paragraph of p. 11 of the revised manuscript: "…more likely to be scavenged by the large incipient soot. For example, the collision frequency function of a naphthalene molecule is approximately 1.6 times smaller than that of an acetylene molecule, based on the kinetic theory of gases.".

The consumption rates of $C_{1-5}$ and $C_{6-10}$ molecules are also obtained by the present ReaxFF simulations, using the number concentration of these classes of molecules at 1350 K within the timeframe of 5.2 (corresponding to the onset of surface growth) to 6.5 ns in Figure 3. In practice, not all molecules are equally reactive, so a collision efficiency, $\alpha$, can be introduced in the equation of the rate of change of the number concentration of the reacting species:

$$\frac{dN}{dt} = -\frac{1}{2}\alpha\beta N^2$$

(R2)

Solving for the differential equation R2, assuming a constant collision rate, $\beta$, yields:

$$a = \frac{2\left(\dfrac{1}{N(t_2)} - \dfrac{1}{N(t_1)}\right)}{\beta\Delta t}$$

(R3)

Equation R3 can be used to provide an estimate of the collision efficiency of $C_{1-5}$ and $C_{6-10}$ species. When applied within the timeframe $\Delta t = 6.5 - 5.2 = 1.3$ ns, it gives:

$$a = \frac{2\left(\dfrac{1}{318} - \dfrac{1}{433}\right)\dfrac{(75.6 \cdot 10^{-10})^3\,m^3}{particles}}{3.58 \cdot 10^{-16}\,{m^3}\big/{s} \cdot 1.3 \cdot 10^{-9}\,s} = 0.00155 \quad \text{for } C_{1-5}$$

(R4)

and

$$a = \frac{2\left(\dfrac{1}{14} - \dfrac{1}{40}\right)\dfrac{(75.6 \cdot 10^{-10})^3\,m^3}{particles}}{5.7 \cdot 10^{-16}\,{m^3}\big/{s} \cdot 1.3 \cdot 10^{-9}\,s} = 0.054 \quad \text{for } C_{6-10} \text{ species}$$

(R5)

These estimated collisions efficiencies indicate that successful reactive collisions are more probable for larger species, also contributing to their faster consumption, even though accurate determination of the collision efficiency would require a trajectory calculation-collision rate theory method approach, similar to that employed in Yang et al. (Yang, Goudeli and Hogan 2018) and Goudeli et al. (Goudeli, Lee and Hogan 2020), to account for reactivity and chemical stability of individual reactive species. The trend observed in the collision efficiencies of equations R4 and R5 is consistent with collision efficiency measurements (Raj et al. 2010) and simulations of PAH dimerization (Totton, Misquitta and Kraft 2012) even though growth by chemical reactions had not been considered in the literature.

To address this Comment, the following sentence was added in the first paragraph of p. 11 of the revised manuscript: "It should be noted, however, that the reactivity and chemical stability of each species vary depending on their size and chemical nature, which also affects their consumption rate.".

*4. P 17: "cyclopropyne (Saxe and Schaefer 1980)" Is this molecule real or a ReaxFF artefact? According to Saxe and Schefer, the singlet state is unstable and the triplet state is metastable at a quite high energy. https://en.wikipedia.org/wiki/Cycloalkyne says: "There is little experimental evidence supporting the existence of cyclobutyne (C4H4) or cyclopropyne (C3H2), aside from studies reporting the isolation of an osmium complex with cyclobutyne ligands.[4]". A few lines later "... indicating they are relatively stable in nature." Is that a valid conclusion? I would say the results indicate that they (cyclopropyne molecules) are stable at the ReaxFF level, but not necessarily in nature.*

4. The Reviewer is correct as cyclopropyne is inherently unstable, and may exist only transiently in high-energy states, making its experimental isolation challenging (Zanda et al. 2020). On the other hand, ReaxFF does not model different spin states. It only provides an approximation of the potential energy surface based on the parametrization and may not accurately predict or describe behaviors specific to higher energy or alternative spin states like the triplet state of cyclopropyne. For this, quantum mechanical methods, such as DFT might be needed to more accurately describe the stability of reactive intermediates.

   To address this, we have rephrased the relevant discussion in the first paragraph of Section 3.2 as follows: "Some of these 3-member rings reported in Table 3 correspond to known monocyclic molecules, including methylene cyclopropane (Zádor, Fellows and Miller 2017) (molecule A), cyclopropyne (Saxe and Schaefer 1980) (molecule B), and cyclopropane (Li, Zhang and Shi 2020) (molecule C), and bicyclic molecules, including bicyclo[1.1.0]butane (Fawcett 2020) (molecule D), spiropentadiene (Billups and Haley 1991) (molecule E) and cyclopropylidenecyclopropane (Güney et al. 2013) (molecule F). It should be noted that some of the 3-member ring structures predicted by ReaxFF, such as cyclopropyne, are not inherently stable and may exist only transiently in high energy states. $C_2H$ cyclic molecules (highlighted in pink) having three-center two-electron configuration (Lammertsma and Ohwada 1996) are observed at all six timesteps ."

*5. P 22: "... allowing for a broader and more accurate exploration of the pathways" I agree to "broader", but I am sceptical regarding "more accurate" without QM or exp validation, see my previous comments.*

5. We agree with the Reviewer and we have rephrased the Conclusions as follows: "These simulations do not assume any specific soot precursors, allowing for a broader exploration of the pathways involved in soot nucleation.".

*6. P 23: "Tracking the pathway of formation of these species could reveal new detailed chemical routes that occur during soot nucleation, which might have not been extensively considered in existing kinetic models, thereby expanding the current understanding of soot formation pathways." I think this would be really useful. Can this be added to the current work or at least the simulation (raw) data provided that are needed for such an analysis?*

6. We agree with the Reviewer. However, as discussed in response to Comment A2, such an analysis requires very frequent recording of both geometry and atomic connectivity data, 250,000 times more frequent than the timeframes used here.

***Technical corrections:***

*P 3: "that shoot growth"*
Corrected.

[revised manuscript text omitted]

Solà, M. (2022) Aromaticity rules. *Nature Chemistry,* 14**,** 585-590.

Totton, T. S., A. J. Misquitta & M. Kraft (2012) A quantitative study of the clustering of polycyclic aromatic hydrocarbons at high temperatures. *Physical Chemistry Chemical Physics,* 14**,** 4081-4094.

Yang, H., E. Goudeli & C. J. Hogan, Jr. (2018) Condensation and dissociation rates for gas phase metal clusters from molecular dynamics trajectory calculations. *The Journal of Chemical Physics,* 148**,** 164304.

Yuan, H., W. Kong, F. Liu & D. Chen (2019) Study on soot nucleation and growth from PAHs and some reactive species at flame temperatures by ReaxFF molecular dynamics. *Chemical Engineering Science,* 195**,** 748-757.

Zádor, J., M. D. Fellows & J. A. Miller (2017) Initiation Reactions in Acetylene Pyrolysis. *The Journal of Physical Chemistry A,* 121**,** 4203-4217.

Zanda, M., R. Bucci, N. L. Sloan & L. Topping (2020) Highly Strained Unsaturated Carbocycles. *European Journal of Organic Chemistry,* 2020**,** 5278-5291.

---

## Author Comment (AC2)

**Reviewer 2**

*The authors presented a reactive MD study of soot inception and early surface growth during acetylene pyrolysis at four temperatures between 1350 and 1800 K relevant to typical hydrocarbon flames. The objective of this study is to provide a more general view of soot inception by investigating the evolution of a reaction system containing C2H2 molecules, rather than assuming some intermediate species or PAHs as the initial condition.*

*1. Overall, the analysis and discussion of the results are comprehensive and offer useful insights into chemical evolution of the simulation system associated with soot inception and early surface growth. However, it is somewhat disappointed to notice that the authors did not provide the details of soot inception, such as the precursor species involved, even though their results should be able to provide such details by examining the numerical results slightly prior to the occurrence of the soot particle shown in Fig. 1.*

1. This is probably a misunderstanding. In Fig. 5 of the original manuscript, we have provided the temporal evolution of $C_2$, C, $C_2H_4$, $C_4H_2$, $CH_4$, $CH_3$, $C_2H_3$, $C_2H$, $C_2H_6$, and $C_2H_5$, which correspond to the most abundant hydrocarbon species at the onset of surface growth. There, i.e., at 0.8 ns for 1650 and 1800 K, at 3.65 ns for 1500 K and at 5.2 ns for 1350 K, the amount of $C_2H_4$, $C_2H_3$, and $C_2H$ molecules peaks, before it drops at 1 ns for 1650 and 1800 K, 4 ns for 1500 K and 5.5 ns respectively after the onset of surface growth, indicating that these species may significantly contribute to the growth of incipient soot nanoparticle but not necessarily to the inception process.

      To further address the Reviewer's Comment, we have discussed the importance of specific precursor species in the soot formation pathways in the end of p. 14 and in p. 15 of the revised manuscript: "The concentration of all molecules increases while $C_2H_2$ is rapidly consumed (Fig. 3a). At the onset of surface growth, $C_2H_3$ exhibits the highest concentration (Fig. 5d, solid line), suggesting that the intermediates formed by reactive collisions of acetylene are eventually converted to vinyl radicals, as proposed in the hydrogen abstraction vinyl acetylene addition (HAVA) mechanism (Shukla 2012).

      For example, at $T$ = 1350 K, the $C_2H_3$ (Fig. 5d) and $C_2H_4$ exhibit a peak (Fig. 5b) at $t$ = 5.25 ns, which corresponds to the timestep right after the onset of the surface growth (Fig. 3). There ($t$ = 5.25 ns), only a small amount of $C_2H$ (Fig. 5d), $C_4H_2$ (Fig. 5b), $C_2$, C (Fig. 5a), and $H_2$ (Fig. 5f) is produced, followed by an increase in $CH_3$ and $CH_4$ (Fig. 5c) at $t \geq 5.5$ ns, indicating that the primary products during acetylene pyrolysis (such as $C_2H_3$, $C_2H_4$) also contribute to the formation of hydrocarbon molecules containing > 5 C. The role of $C_2H_3$ has also been highlighted in kinetic mechanisms as a key contributing molecule to the formation of cyclopentadiene through its addition to $C_4H_6$, which competes with benzene formation pathways and influences the composition and growth of soot particles (Faravelli, Goldaniga and Ranzi 1998). The amount of $C_2H_4$ (Fig. 5b: solid lines), $CH_4$ (Fig. 5c: solid lines), $C_2H_6$ (Fig. 5e: solid lines), and $H_2$ (Fig. 5f: solid lines) hardly changes at longer times, after $C_2H_2$ has practically been depleted, indicating that these species do not contribute to the growth of the soot cluster. The abrupt decrease in $C_2H_4$ (Fig. 5b: solid lines), $C_2H_3$ (Fig. 5d: solid lines) and $C_2H$ concentration (Fig. 5b: broken lines) also hints that these species contribute towards the growth of incipient soot. It should be noted that the contribution of polyynes, such as $C_4H_2$ and $C_2H$, has also been recognized in accelerating polymerization reactions that lead to soot nucleation (Indarto 2008)."

      In addition, the detailed list of molecules and their chemical structure slightly prior to the occurrence of the soot nanoparticle has been provided for $T$ = 1350 K in Table S4 at $t$ = 5 ns (i.e., 0.2 ns prior to the onset of surface growth), as well as for $T$ = 1650 and 1800 K in Tables S7 and S12, respectively, at $t$ = 0.8 ns (i.e., 0.05 ns prior to the onset of surface growth). To clarify this, the 1st and 2nd paragraph in p. 21 of the revised manuscript have been modified as follows: "… (Supporting Information: Table S3). For 1350 K at $t$ = 5 ns (Supporting Information: Table S4), 3-member rings and molecules containing > 5 C atoms are formed, coinciding with $C_2H_2$ consumption (Fig. 3a). At this temperature, hardly any 5- and 6-member rings are found in the reported chemical structures (Supporting Information: Tables S3-S6), which exclude the incipient soot, indicating that most of these rings are formed within the incipient soot nanoparticle. Nevertheless, at $t$ = 5 ns (i.e., 0.2 ns prior to the onset of surface growth), a molecule with a 6-member ring appears in the $C_6$-$C_{10}$ range (Supporting Information: Table S4), but most of the larger molecules with > $C_{10}$ are primarily long aliphatic chains.

      Smaller benzene derivatives are observed in the reaction pathway at 1650 (Tables S10-S11) and 1800 K

(Tables S14-S15). At both 1650 (Tables S7-S11) and 1800 K (Tables S12-S15), mostly 3-member rings are formed up to 6 ns along with the incipient soot (excluded in the tables). Approximately 0.05 ns prior to the onset of surface growth (Table S12: $t$ = 0.75 ns), a 5-member ring spontaneously forms. At $T$ = 1650 K,…".

*2. The authors paid close attention to the evolution of 3-, 5-, and 6-member rings in their results, but did not mention if there are other polycyclic aromatic species in the system, such as 4- or 7-member rings.*

2. We thank the Reviewer for this Comment. To address this, we have quantified the temporal evolution of the number of 4- and 7-member rings at various temperatures. Figure R1 shows that a negligible number of 4-member rings is formed transiently. These structures are rather unstable and practically disappear over time. In contrast, a considerable number of 7-member rings is formed, comparable to that of 5-member rings (Fig. 6b). All 7-member rings appear after the onset of surface growth and belong to the incipient soot nanoparticle.

Figure R1 has been inserted as Figure S16 in the revised Supporting Information. In addition, the following sentence has been added in the end of Section 2.3 of the revised manuscript: "The 4- and 7-member ring structures are determined by MAFIA-MD (Mukut, Roy and Goudeli 2022)." and the following clauses have been included in the paragraph in p. 16: "…and 1800 K (stars), with 4- and 7-member rings shown in Fig. S16. The …" and "…before the onset of surface growth, no 5-, 6- and 7-member rings are observed (Fig. 6 and S16), indicating…".

[Figure]

**Figure R1.** Temporal evolution of (a) 4- and (b) 7-member rings at 1350 (circles), 1500 (squares), 1650 (diamonds), and 1800 K (stars). A negligible number of 4-member rings is formed transiently, indicating that these structures are rather unstable and practically disappear over time. In contrast, a considerable number of 7-member rings is formed, comparable to that of 5-member rings (Fig. 6b). All 7-member rings appear after the onset of surface growth and belong to the incipient soot nanoparticle.

*3. The authors also noticed that their results are somewhat different from those reported in the literature for different fuels and at different temperatures (Wang et al., 2022a, Wang et al., 2022b, Zhao et al., 2020). Can the authors speculate if the chemical details of soot formation and early growth will be different for different hydrocarbons?*

3. We thank the Reviewer for raising this question. Indeed, soot formation pathways may differ depending on the fuel type, and nucleation conditions such as temperature and pressure (Richter and Howard 2000, Reizer, Viskolcz and Fiser 2022). Even though the detailed chemical structure of soot and its precursors are scarcely provided, to the best of our knowledge, the following qualitative insights can be obtained based on the literature:

The bulk ReaxFF literature focuses on high temperature conditions (>2400 K) that are more relevant to carbon black formation rather than to soot. For example, formation of carbonaceous nanoparticles during acetylene pyrolysis at high temperature (3000 K) has been explored by ReaxFF (Zhang et al. 2015) using the same forcefield as in the present study. The clusters and larger nanoparticles reported in this study (Fig. R2) resemble those obtained in our work (Fig. 1 in the revised manuscript), with large linear-like chains occupying the surface of the nanoparticle while islands of carbon rings occupy the cluster core (Fig. R2d-g).

[Figure]

**Figure R2.** Typical structures showing the stages in carbon black formation (adjusted from (Zhang et al. 2015).

However, fuels such as n-decane and n-heptane, lead to compact onion-like soot nanoparticles with higher ring content and shorter aliphatic side chains compared to soot structures obtained by acetylene pyrolysis at the same temperature range (Fig. R2). For example, pyrolysis of n-decane at 3000 K (Liu et al. 2020) has revealed the formation of PAH-like structures (Fig. R3). During thermal degradation at 3000 K, n-decane decomposes into smaller molecules such as $CH_4$, $C_2H_4$, and $C_2H_3$, consistent with those observed in the present work (Fig. 5). Reactive collisions gradually result in PAH-like molecules and larger soot nanoparticles with graphitic-like structure. These results are consistent with those obtained by n-heptane pyrolysis at $T = 2200 – 2600$ K in Fig. R4 (Fakharnezhad et al. 2025), indicating that the fuel type can significantly affect the formation mechanism and structure of soot.

[Figure]

**Figure R3.** Snapshots during soot formation by n-decane pyrolysis at 3000 K (adjusted from (Liu et al. 2020)).

[Figure]

**Figure R4.** Snapshots of soot formation by n-heptane pyrolysis at 2500 K (Fakharnezhad et al. 2025).

To clarify this, we have modified the end of the last paragraph of p. 21 of the revised manuscript as follows: "…It should also be noted that at temperatures below 1800 K, which are relevant to soot formation in flames, the ReaxFF-predicted cluster structures are significantly different from those formed by acetylene combustion at 2700 K (Wang et al. 2022a), by methane and ethylene combustion at 3000 K (Wang et al. 2022b), by dimerization of PAHs (Zhao et al. 2020), and by pyrolysis of n-decane at 3000 K (Liu et al. 2020) n-heptane at 2200 – 2600 K (Fakharnezhad et al. 2025), where PAH-like soot precursors are formed leading to large carbonaceous clusters mainly composed of rings and containing only a small fraction of branches. This indicates that the fuel type along with the nucleation conditions can significantly affect the formation mechanism and structure of soot.".

*Minor points:*

*• In line 17 on page 3, 'shoot' should be soot.*

Corrected.

*• In the line below Fig. 5, why do you call the results in Fig. 6 'the cumulative number …' rather than 'the number …'?*

Thank you. We have replaced "the cumulative number" with "the number".

*• In the first paragraph on page 16, Figure S15a and Figure 15b seem messed up.*

Thank you. We corrected the caption of Figure S15 of the original manuscript (Figure S17 in the revised Supplementary Information).

**References**

Fakharnezhad, A., D. M. Saad, G. A. Kelesidis & E. Goudeli (2025) Nucleation Rate of Soot by n-Heptane Pyrolysis. *Aerosol Science & Technology,* accepted.

Faravelli, T., A. Goldaniga & E. Ranzi (1998) The kinetic modeling of soot precursors in ethylene flames. *Symposium (International) on Combustion,* 27**,** 1489-1495.

Indarto, A. (2008) Soot Growth Mechanisms from Polyynes. *Environmental Engineering Science,* 26**,** 1685-1691.

Liu, L., H. Xu, Q. Zhu, H. Ren & X. Li (2020) Soot formation of n-decane pyrolysis: A mechanistic view from ReaxFF molecular dynamics simulation. *Chemical Physics Letters,* 760**,** 137983.

Mukut, K. M., S. Roy & E. Goudeli (2022) Molecular arrangement and fringe identification and analysis from molecular dynamics (MAFIA-MD): A tool for analyzing the molecular structures formed during reactive molecular dynamics simulation of hydrocarbons. *Computer Physics Communications,* 276**,** 108325.

Reizer, E., B. Viskolcz & B. Fiser (2022) Formation and growth mechanisms of polycyclic aromatic hydrocarbons: A mini-review. *Chemosphere,* 291**,** 132793.

Richter, H. & J. B. Howard (2000) Formation of polycyclic aromatic hydrocarbons and their growth to soot—a review of chemical reaction pathways. *Progress in Energy and Combustion Science,* 26**,** 565-608.

Shukla, B. a. K., Mitsuo (2012) A novel route for PAH growth in HACA based mechanisms. *Combustion and Flame,* 159**,** 3589-3596.

Wang, Y., M. Gu, D. Liu & X. Huang (2022a) Soot growth mechanism in C2H2 combustion with H2 addition: A reactive molecular dynamics study. *International Journal of Hydrogen Energy*.

Wang, Y., M. Gu, Y. Zhu, L. Cao, J. Wu, Y. Lin & X. Huang (2022b) Analysis of soot formation of CH4 and C2H4 with H2 addition via ReaxFF molecular dynamics and pyrolysis–gas chromatography/mass spectrometry. *Journal of the Energy Institute,* 100**,** 177-188.

Zhang, C., C. Zhang, Y. Ma & X. Xue (2015) Imaging the C black formation by acetylene pyrolysis with molecular reactive force field simulations. *Physical Chemistry Chemical Physics,* 17**,** 11469-11480.

Zhao, J., Y. Lin, K. Huang, M. Gu, K. Lu, P. Chen, Y. Wang & B. Zhu (2020) Study on soot evolution under different hydrogen addition conditions at high temperature by ReaxFF molecular dynamics. *Fuel,* 262**,** 116677.